# T2Candida for the Diagnosis and Management of Invasive *Candida* Infections

**DOI:** 10.3390/jof7030178

**Published:** 2021-03-03

**Authors:** Lea M Monday, Tommy Parraga Acosta, George Alangaden

**Affiliations:** 1Division of Infectious Diseases, Department of Internal Medicine, Henry Ford Hospital, Detroit, MI 48202, USA; lmonday1@hfhs.org (L.MM.); tparrag1@hfhs.org (T.P.A.); 2School of Medicine, Wayne State University, Detroit, MI 48202, USA

**Keywords:** T2Candida, candidemia, invasive candidiasis, fungemia

## Abstract

Invasive candidiasis is a common healthcare-associated infection with high mortality and is difficult to diagnose due to nonspecific symptoms and limitations of culture based diagnostic methods. T2Candida, based on T2 magnetic resonance technology, is FDA approved for the diagnosis of candidemia and can rapidly detect the five most commonly isolated *Candida* sp. in approximately 5 h directly from whole blood. We discuss the preclinical and clinical studies of T2Candida for the diagnosis of candidemia and review the current literature on its use in deep-seated candidiasis, its role in patient management and prognosis, clinical utility in unique populations and non-blood specimens, and as an antifungal stewardship tool. Lastly, we summarize the strengths and limitations of this promising nonculture-based diagnostic test.

## 1. Introduction

*Candida* is the most common cause of invasive fungal infections and associated mortality in the United States [1]. Invasive candidiasis includes both candidemia and deep-seated *Candida* infections at submucosal sites that may occur with or without associated bloodstream infection. The prevalence of invasive candidiasis has steadily increased in a broad variety of clinical settings. The increase is in part due increased survival with advances in life support techniques in the intensive care unit (ICU), expanding use of therapeutic modalities including immunosuppressive therapies, complex surgical procedures, hematopoietic stem cell and solid organ transplantation and others. Between 1970 and 2000, the annual number of sepsis cases due to fungal organisms increased by 207% in the United States [2]. Invasive candidiasis has significant attributable mortality and early antifungal treatment is associated improved outcomes [3,4]. Despite recognition of risk factors, the diagnosis and consequently treatment of invasive candidiasis is frequently delayed. Cultures of blood and specimens from deep-seated sites of infection have sensitivity of approximately 50% and slow turnaround times [5,6,7]. Furthermore, blood cultures will not identify deep-seated invasive candidiasis unless it is associated with candidemia.

These diagnostic limitations leave clinicians struggling to balance the benefits of early empiric antifungal therapy (EAT) with the risks of antifungal resistance and healthcare costs associated with unnecessary antifungal use. Rapid and sensitive non-culture based diagnostic methods for detecting invasive candidiasis are needed to improve outcomes through early initiation of targeted antifungal therapy [5,6,7]. In this review, we discuss T2Candida, the novel technology of this automated non-culture based diagnostic platform using whole blood specimens, and preclinical and clinical studies that evaluated the performance characteristics of this test for detection of candidemia. We also examine the role of T2Candida for the diagnosis of deep-seated candidiasis (with and without candidemia), its potential as a prognostic and management tool for invasive candidiasis, and its use as a stewardship resource to de-escalate antifungal therapy. Lastly, we discuss the clinical utility of this test in unique patient populations and testing of non-blood clinical specimens.

## 2. T2 Candida Panel

T2Candida is a non-culture-based platform approved by the US Food and Drug Administration (FDA) in 2014 for the diagnosis of candidemia [8]. Whole blood specimens are collected in K2EDTA tubes and inserted into the fully automated T2Dx instrument (T2Biosystems, Inc., Wilmington, MA, USA). T2Dx lyses the *Candida* cells by mechanical bead beating then amplifies their DNA using a thermostable DNA polymerase and primers for the *Candida* ribosomal DNA operon [9,10]. The amplified *Candida* DNA product is detected using amplicon-induced agglomeration of super magnetic particles and T2 Magnetic Resonance (T2MR) measurement [9]. An internal control is processed with each specimen to monitor the integrity of the results. The resulting product is reported as positive or negative for identification of the 5 common *Candida* species (*C. albicans, C. tropicalis, C. parapsilosis, C. krusei, and C. glabrata*) which account for >95% of candidemia at most centers [9,10,11,12]. The results are grouped on the basis of susceptibility to antifungals (primarily fluconazole) and are reported as *C. albicans/C. tropicalis, C. parapsilosis, and C. krusei/C. glabrata*). The T2Candida Panel has a limit of detection as low as 1 colony-forming unit (CFU)/mL of whole blood and has a mean turn-around-time of < 5 h [8,9,10,11,12]. 

## 3. Performance of T2Candida: Preclinical and Early Clinical Trials

Initial performance was evaluated in two preclinical studies using blood samples spiked with the 5 *Candida* sp. detected by T2Candida namely *C. albicans/C. tropicalis*, *C. parapsilosis*, and *C. krusei/C. glabrata*) [9,10]. In one early study, T2Candida was compared to BACTEC^®^ automated blood culture system (Becton Dickinson and Company, Franklin Lake, NJ, USA) for detection of *Candida* sp. [9]. By day 5, the automated blood culture system detected *Candida* growth in 100% of the bottles for all *Candida* sp. except *C. glabrata* where no growth (0%) was detected in any of the 20 bottles tested. In comparison T2Candida had a 100% detection rate for all *Candida* sp. including *C. glabrata* [9]. The T2Candida assay had sensitivity of 100% and specificity of 96 to 100% (Table 1). The average time to detection was significantly shorter for the T2Candida panel (3–5 h assay time) compared to 63 h for the automated blood culture system (Table 2) [9]. Comparative testing using 133 spiked blood samples demonstrated a high concordance between T2Candida panel and automated blood cultures with 98% positive agreement and 100% negative agreement (Table 3) [10]. T2candida had an overall limit of detection of 1–3 CFU/mL for the 5 *Candida* sp. (Table 4) [10]. The increasing use of empiric antifungal therapy prior to obtaining blood samples can impact the accuracy of diagnostic tests. The performance of T2Candida and blood cultures to detect *Candida* in presence of antifungals was evaluated using whole blood samples spiked with *Candida* sp. with or without fluconazole and caspofungin [13]. Neither antifungal impacted the performance of the T2Candida while fluconazole reduced the overall blood culture sensitivity by 7.5% at the low inoculum and by 12.5% at the high inoculum and prolonged the time to detection [13].

Early clinical performance was evaluated in two large multicenter trials named, “Detecting infections rapidly and easily for candidemia trials” (DIRECT and DIRECT2) [14,15]. In the DIRECT trial, whole blood samples for T2Candida testing were obtained from 1801 hospitalized patients that had blood cultures ordered for routine standard of care. In total, 250 blood samples were manually supplemented with the 5 *Candida* species in concentrations ranging from <1 to 100 CFU/mL.14 Mean time to *Candida* species detection was 4.4 ± 1 h with a sensitivity and specificity of 91% and 98%, respectively, (Table 1, Table 2, Table 3 and Table 4). DIRECT2 enrolled 152 patients with candidemia documented on blood cultures with any of the five *Candida* sp. detected by T2Candida. Follow-up blood cultures and concurrent whole blood samples for T2Candida testing were obtained after enrollment (Table 5) [15]. Overall T2Candida sensitivity was 89% and the test was more likely to be positive than follow-up blood cultures (45% versus 24%), with the strongest association in patients who were neutropenic or receiving prior antifungal therapy [15]. Efficacy of T2Candida in monitoring candidemia clearance compared to blood culture was further evaluated in the “Serial therapeutic and antifungal monitoring protocol” (STAMP) trial [16]. In this multicenter clinical trial of 31 patients with *Candida* sp. isolated from their pre-enrollment blood culture, 13 (42%) had at least 1 positive post-enrollment surveillance blood cultures or T2Candida positive result. In the 93 sets of blood cultures and T2Candida specimens collected, 7.5% blood cultures were positive compared to 25% of T2Candida specimens. All positive surveillance blood cultures had a concordant positive accompanying T2Candida result for the same *Candida* sp., but only 7 of the 23 positive T2Candida were detected by blood culture. These results suggest T2Candida may be better than blood cultures for monitoring the clearance of candidemia in patients receiving antifungal therapy. 

Based on data from the DIRECT1 [14] and DIRECT2 [15] trials, Clancy and Nguyen estimated the performance of T2Candida and positive and negative predictive values (PPVs/NPVs) in various patient populations at high-risk for candidemia [17]. The PPV was heavily dependent on pre-test probability (and therefore prevalence) in the patient population being tested, however, the NPV remained high (98–100%) across all prevalence values ranging from <1% to 10% [17]. A subsequent meta-analysis by Tang et al. [18], evaluated the performance of T2Candida in 2,717 research subjects from 8 published articles (six manuscripts including DIRECT1 [14], DIRECT2 [15], and STAMP [16], and two published abstracts) [18]. The authors found a pooled sensitivity of 91% (95% confidence interval (CI): 0.88–0.94) and specificity 94% (95% CI: 0.93–0.95) [18].

## 4. T2Candida as a Prognostic Indicator for Patients with Known or Suspected Candidemia

Two studies in Spain were among the first to explore the role of T2Candida as a prognostic tool [19,20]. In a prospective multicenter study in Madrid, 30 patients with known candidemia had serial follow up blood cultures, T2Candida, and B-d-glucan (BDG) at 5 subsequent time points over 0 to +14 days [19]. A positive T2Candida result within the first 5 days of blood culture positivity was associated with a 37-fold higher rate of complicated candidemia (defined as death attributable to candidemia, or metastatic deep-seated infection). T2Candida was a better marker for predicting risk for complicated infection than serial blood culture or BDG [19]. A second prospective observational study by the same authors evaluated the potential role of T2Candida, *Candida albicans* germ tube antibody (CAGTA), and BDG for predicting poor outcome in patients initiated on empiric antifungals for suspected invasive candidiasis [20]. CAGTA, BDG, and T2Candida were obtained at baseline, +2, and +4 days in 49 enrolled patients of whom 14 had a poor outcome (defined as death or a proven diagnosis of invasive candidiasis in the first 7 days). Positive baseline T2Candida result was independently associated with poor outcome (PPV 100% and NPV 79.6%) while BDG and CAGTA were not predictive. A post hoc analysis of 32 patients with known candidemia from the DIRECT2 [15] trial, examined the association of a positive T2Candida result and 28-day mortality. In these patients (69% were on antifungal therapy at the time of testing) mortality was 40% if T2Candida was positive and 9% if negative (*P*=0.06) [21]. Taken together, mortality was 42% if either T2Candida or blood culture were positive and 5% if both were negative (*P*=0.02) [21]. These studies indicate that T2Candida in combination with blood cultures provide important prognostic information in patients with suspected invasive candidiasis on empiric and definitive antifungal therapy.

## 5. T2Candida for the Diagnosis of Deep-Seated Candida Infection

Invasive candidiasis as a disease spectrum includes not only candidemia (with or without deep-seated infection), but also isolated deep-seated candidiasis without candidemia [5]. Deep-seated *Candida* infection is difficult to study because it includes a heterogenous group of conditions ranging from direct local inoculation (e.g., *Candida* peritonitis in peritoneal dialysis, spinal infection from contaminated steroid injections), to hematogenous metastatic seeding of *Candida* sp. (e.g., infective endocarditis, hematogenous fungal endophthalmitis) [22]. Intra-abdominal candidiasis (IAC) usually refers to peritonitis and abdominal abscess in patients with recent abdominal surgery, anastomotic leak, or gastrointestinal perforation [22,23,24]. 

The usefulness of positive T2Candida results despite negative blood cultures for the identification of IAC has been noted in a few cases. These included patients with IAC and positive T2Candida result concordant with *Candida* sp. isolated from cultures of peritoneal fluid, infected bile16, and explanted liver tissue [25]. Building on these preliminary case reports, a 2019 ICU study examined T2Candida performance in a real-world patient population with suspected invasive candidiasis including IAC [26]. Blood cultures, T2Candida, and *Candida* Mannan antigen (MAg) were performed on 126 ICU patients at high-risk for invasive candidiasis with sepsis despite 3 days of empiric broad-spectrum antibiotics. Paired samples were obtained twice weekly (334 sets) and patients were classified as having proven, likely, or possible invasive candidiasis based on culture results, imaging, and expert review [26]. At enrollment, 77% were on antifungal therapy. In all, 28 of 126 (22%) had proven, likely, or possible invasive candidiasis and IAC was the most common manifestation in 57% of these cases. Five of the 11 proven cases had *Candida* sp. isolated on blood cultures and T2Candida detected all except a *C. kefyr* that is not included on the panel. Overall, the sensitivity of T2Candida, blood culture and MAg in proven cases of invasive candidiasis was 55%, 45% and 36% [26]. The addition of T2Candida to blood culture or MAg resulted in the best diagnostic performance. All three tests had negative predictive value of >90% [26]. A later study compared T2Candida with blood cultures and cultures from deep-seated infection [27]. In total, 133 samples were taken from 32 patients with candidemia and 22 patients with deep-seated invasive culture proven candidiasis. T2Candida was positive in 27% of the patients with deep-seated invasive candidiasis [27]. Of note, only 88% patients with candidemia had positive T2Candida result at any time point, which is lower than rates reported in the early clinical trials [27]. Lastly, a 2020 study compared performance of T2Candida to BDG and blood cultures for the diagnosis IAC [28]. Patients were enrolled if they were at risk for IAC based on recent gastrointestinal tract perforation, necrotizing pancreatitis, or abdominal surgery as well as concurrent *Candida* colonization from a nonsterile site. Of 48 patients enrolled, 38% had proven IAC defined as *Candida* isolated by perioperative intra-abdominal culture [28]. In patients with proven IAC, blood cultures and T2Candida were positive in 11% and 33%, respectively [28]. Two patients had IAC with species not included on the T2Candida panel (*C. kefyr* and *C. lusitaniae*) [28]. Overall, T2Candida sensitivity/specificity were 33%/93% and PPV/NPV 71%/74% for diagnosing IAC. The above studies provide limited data that T2Candida may serve as a surrogate marker for deep-seated candidiasis and IAC, especially in cases with negative blood cultures or when invasive procedures cannot be performed to obtain specimens for testing from deep-seated sites of infection.

## 6. T2Candida Cost-Effectiveness and Potential Role in Antifungal Stewardship

Antifungals, like antimicrobials, require active stewardship initiatives to ensure responsible use and minimize development of resistance. The recent guidance from the Mycoses Study Group for antifungal stewardship recommend that all centers frequently managing patients with invasive fungal infection have access to timely conventional and non-culture based diagnostic methods for *Candida* species [29]. However, diagnostic stewardship (responsible use of the diagnostic test itself), is an important consideration especially for newer more costly non-culture-based methods such as T2Candida. Several studies have investigated the utility of T2Candida to de-escalate antifungals or lead to cost savings. 

Two early publications reported on the cost-effectiveness of T2Candida based on theoretical decision-tree models [30,31]. The first study estimated that a hospital with 5100 annual high-risk invasive candidiasis patients could possibly save $5.8 million dollars and avert 61% in theoretical candidemia-related mortality [30]. A second publication used a decision-tree model to compare the cost-effectiveness of T2Candida to that of empiric antifungal therapy (EAT) or blood culture directed therapy (BCDT) [31]. T2Candida was estimated to be less expensive and more effective than BCDT, but less expensive and less effective than EAT with an echinocandin [31]. A sensitivity analysis showed that cost effectiveness was highly dependent on the prevalence of candidemia and the greatest benefit would be in the ability to withhold or stop empiric therapy in low-risk patients [31]. Both studies are limited by their theoretical nature and industry sponsorship. In real-life true cost savings may vary based given frequent changes in the pricing of various antifungals. 

In addition, two other pilot studies investigated the potential effects that negative T2Candida results may have on EAT effects in real patient populations [32,33]. In an early study from 2014, authors recorded an average time to yeast identification of 2.2 ± 1.3 days and average time to start antifungal of 3.5 ± 21 days in 162 real patients with candidemia [32]. In subsequent Monte Carlo simulations, the time to initiation of antifungals was reduced to 0.6 ± 0.2 days with T2Candida, 2.6 ± 1.3 days for PNA-FISH (fluorescence in situ hybridization using peptide nucleic acid probes), and 2.5 ± 1.4 days for MALDI-TOF (matrix-assisted laser desorption/ionization time of flight). Assuming EAT is stopped when T2Candida returned negative, they estimated a theoretical reduction of 3136 to 6078 fewer echinocandin doses per 5000 patients annually [32]. A single-center prospective observational pilot study examined the potential effect of T2Candida on EAT in 46 patients at high-risk for candidemia with severe sepsis and receiving empiric therapy with an echinocandin for a median duration of 7 days [33]. Compared to blood cultures, T2Candida reduced time to a negative result by 5 days with a NPV of 100%, providing proof of concept that it may be a reasonable tool to use for stopping EAT.

Real-world experience of the effect of T2Candida on antifungal use has been investigated in 4 observational studies [34,35,36,37]. Patch et al. [34] examined the effect of T2Candida implementation in a multi-hospital community health system on time to initiation of antifungal therapy in patients with confirmed candidemia, and utilization of empiric micafungin those with suspected candidemia. The pre-implementation group of 19 patients with candidemia received antifungal therapy an average of 34 hours from blood culture draw, compared to 20 patients in the T2Candida group that received antifungal therapy at an average of 6 h (P = 0.0015) [34]. Despite the more rapid initiation of targeted antifungals, there was no significant difference in length of stay (LOS), all-cause 30-day readmissions, or mortality. The average duration of therapy (DOT) with micafungin despite a lack of mycological evidence of infection in the pre-T2Candida phase was 6.7 days compared with 2.4 days in the T2Candida post-implementation cohort. This resulted in total savings of $280 per patient tested due to reduced unnecessary antifungal costs [34]. In 2017, Wilson et al. published a quasi-experimental study examining the effects on time to appropriate antifungal, time to candidemia detection, and patient outcomes before and after implementation of T2Candida in a 4-hospital academic Michigan health system [35]. Among 161 patients with probable or proven candidemia, the overall median time to appropriate therapy was reduced from 39 to 22 h (*p* = 0.003). In the subgroup of 37 of the 74 post-implementation phase patients with positive T2Candida, the median time to directed antifungal therapy was 5 h [35]. Interestingly, ocular candidiasis was identified on ophthalmology examination in 30% of the pre-T2Candida cohort compared to 12% in the post-T2Candida cohort [35]. Authors postulated that earlier detection and initiation of antifungals could have contributed to the decreased incidence of eye infection, however, no significant differences in LOS or mortality were noted between groups [35]. A subsequent quasi-experimental study in the same health system compared duration of EAT with anidulafungin in ICU patients with suspected candidemia (sepsis despite 72 h of broad-spectrum antibiotics and *Candida* score ≥3) before and after implementation of T2Candida [36]. In the pre-implementation phase, an algorithm-based care plan was utilized to obtain BDG and blood cultures and initiate EAT, then T2Candida was utilized instead of BDG in in the post-intervention phase [36]. EAT was discontinued if both blood cultures and BDG or T2Candida were negative. In total, 103 patients were included in each group. The median duration of EAT in the BDG group was 2 days compared to 1 day in the T2Candida group (*p* < 0.001) [36]. Development of proven candidemia after discontinuation of EAT was 8% and 3% in the BDG and T2Candida groups, respectively. Inpatient mortality was higher in the T2Candida group, 60% vs. 43% (*p* = 0.018), which was possibly attributable to significantly higher severity of illness based on QSOFA scores and vasopressor usage at baseline in the T2Candida group [36]. Most recently, a study at an Indianapolis academic health center retrospectively evaluated the sensitivity, specificity, PPV, NPV, and antifungal DOT/1000 patient days before after implementation of T2Candida [37]. A total of 433 patients were evaluated, including 16 with positive T2Candida and negative blood cultures and 6 with negative T2Candida and positive blood cultures. Overall sensitivity and specificity were 65% and 96%; with a pre-test likelihood of 4.4%, the PPV was 41% and NPV 99%. The overall fluconazole DOT/1000 patient days before, during T2Candida use, and after removal of T2Candida was 45.3, 41.6, and 47.8 days, respectively (*p* = 0.017 before versus during; *p* = 0.012 during versus after) [37]. The DOT/1000 days of micafungin improved before versus during from 14.2 to 9.9 days (*p* < 0.001), notably, the use of micafungin continued to decline to 8.8 days (*p* = 0.252 during versus after) with the removal of T2Candida [37]. Patient outcome data were not collected, and two additional infectious disease pharmacists actively provided audit and feedback which may have contributed to continued reduction in use of micafungin [37]. 

These studies suggest the use of T2Candida in conjunction with blood cultures can facilitate and enhance antifungal stewardship efforts in hospitalized patient populations at high-risk for candidemia and invasive candidiasis. 

## 7. T2Candida in Other Patient Populations and Non-Blood Specimens

A single study investigated the use of T2Candida to diagnose candidemia in pediatric patients [38]. Authors developed a method to reduce the amount of blood required by the T2Candida system from >3 mL per manufacturer recommendations to only 2 mL by directly pipetting whole blood into the T2Candida cartridge [38]. Fifteen whole blood samples were collected from pediatric patients with known candidemia as well as nine negative controls. T2Candida results were 100% concordant with blood culture results from all 21 patients [38]. Although limited by a small sample size, this represents the only study of T2Candida to diagnose candidemia in the pediatric patients.

Few reports have evaluated T2MR for the detection of *Candida* sp. in non-blood clinical specimens. Kouri et al. examined T2MR for detection of *Candida* in peritoneal dialysate (PD) fluid [39]. Culture negative fluid from three healthy pediatric peritoneal dialysis patients were spiked with *C. glabrata* in serial dilutions of 10-2, 10-4, and 10-6 along with negative controls [39]. PD fluid compositions of 1.5% dextrose, 2.5% dextrose, and a combination of both were included to assess interference over a range of dextrose concentrations. Spiked dialysates were loaded into 4mL tubes and run on the T2MR instrument according to the manufacturer’s instructions for whole blood analysis. All assay results were valid, no interference identified, and there was 100% concordance between spiked specimens and the negative/positive samples [39]. Given that fungal PD fluid cultures typically take 24–48 h to turn positive, the rapid turnaround time of T2MR may represent an opportunity to initiate early targeted antifungals and discontinue unnecessary intraperitoneal antibiotics. Two published abstracts have investigated T2Candida in the setting of ocular candidiasis, a potential sight-threatening condition that may require modification of systemic antifungal therapy or the intravitreal instillation of antifungals [40,41]. A retrospective evaluation was performed of 164 patients that had inpatient ophthalmology consultation for suspected ocular candidiasis at a Detroit academic hospital [40]. Of these, 60% had a positive T2Candida result and 73% had blood culture positive for Candida sp. Concordance or discordance between the two tests was not reported. Ophthalmology examination identified 13% patients with definite ocular candidiasis. Sensitivity of T2Candida was 75% for those with definite chorioretinitis as compared to 64% for blood cultures [40]. Another study reviewed 360 episodes of blood culture proven candidemia and 288 episodes of T2Candida positive alone (without positive blood cultures) over the course of 4 years at an Alabama health system [41]. Of those who underwent dilated fundoscopic examination, ocular candidiasis was present in 13% in the blood culture group and 9% in the T2Candida positive group (*p* = 0.177) [41]. Identification of *C. parapsilosis* was significantly more common in the T2Candida patients, however, there were no differences in presence of visual symptoms, type of ocular involvement, need for intravitreal injection, or mortality [41]. These early studies suggest a potential adjunctive role of T2Candida in the diagnosis and early treatment of ocular candidiasis especially in the setting of negative blood cultures.

## 8. T2Candida Panel for the Detection of *Candida auris*

*Candida auris* is an emerging multidrug resistant pathogenic yeast that can colonize multiple body sites and survive for weeks in the environment facilitating healthcare associated outbreaks [42]. Composite skin swabs used to assess colonization using culture-based methods are sensitive and specific, however, require 14 days which limits the early operativity to isolate patients appropriately. In 2018, Sexton et al. reported the use of diagnostic primers built into the established T2MR platform to identify *C. auris* [43]. Swabs from the axilla and groin were collected from 89 patients and submitted to the Center for Disease Control (CDC) for *C. auris* screening via culture and confirmation by MALDI-TOF MS. The T2Dx instrument requires blood collection vacutainers so 100 µL of unprocessed liquid buffer from the patient swab sample was added to vacutainers and then brought to a final volume of 3mL with buffered saline solution. The T2MR *C. auris* assay recognized isolates from each of the 4 known clades of *C. auris* with a sensitivity of 89% and specificity of 98% [43]. A newly developed T2Cauris panel including species not previously included (*C. auris*, *C. haemulonii*, *C. duobushaemulonii* and *C. lusitaniae*) recently presented by T2Biosystems ^®^ at the IDSA 2017 national conference, was able to rapidly detect *C. auris* in whole blood samples or from common patient and environmental swab matrices [44]. The panel is now available for research purposes [45].

## 9. T2Candida: Strengths and Limitations

T2Candida represents a novel, non-culture, direct from whole blood, diagnostic tool to rapidly identify patients with invasive candidiasis, however there are several limitations. First, sensitivity and PPV are dependent on pre-test probability, so clinicians must use it in appropriate setting where the prevalence of invasive candidiasis is high, e.g., ICU patients with risk factors for invasive candidiasis and sepsis unresponsive to broad spectrum antibiotics [11,17]. A T2Candida and blood culture guided management algorithm for suspected invasive candidiasis in ICU patients used at our institution is shown in Figure 1. In care settings with low pre-test likelihoods of candidemia, PPVs are too low to justify initiating antifungals based purely on a positive result [11,17]. The strong NPVs of T2Candida across a broad range of prevalence’s, one of its strengths is as a stewardship tool to aid in decisions regarding discontinuation of empiric antifungal treatment [17,29]. The standard T2Candida panel currently identifies the five commonly isolated *Candida* sp. and may limit its applicability [26,28]. A recent review of 686 *Candida* blood isolates collected over a 12-year period in Spain found that 91% were those included in the T2Candida panel [46]. In adult units, *C. albicans*, *C. parapsilosis*, and *C. glabrata* accounted for 91–100% of all isolates whereas in pediatric units, *C. albicans* and *C. parapsilosis* were frequent [46]. The selection of centers for T2Candida implementation should account for local epidemiology and distribution of *Candida* sp. It remains unclear how to clinically interpret the approximately 30% of cases that have positive T2Candida with negative blood cultures [22]. Blood cultures can detect 1CFU/mL of most *Candida* sp. and remain the gold standard, hence it is unknown what proportion of the T2Candida positive and blood culture negative results represent false-negative cultures, deep-seated occult infection with negative blood cultures, or false-positive T2Candida results. The quick turn-around-time of T2Candida, ability to rapidly detect potential azole-resistant species, e.g., *C. krusei* and consequent earlier initiation of appropriate antifungal therapy would be expected to produce a noticeable survival benefit. However, improved outcomes have not yet been conclusively demonstrated in observational studies [34,35,36].

A recent multicenter study isolated mixed yeast infections in approximately 2% of cultures from sterile sites [47]. The paired reporting of T2Candida results (i.e., *C. albicans*/ *C. tropicalis* and *C. glabrara*/*C. krusei*) would make accurate identification of mixed yeast infections difficult. Lastly, T2Candida results could have impacted the reporting of central line-associated-bloodstream infection (CLABSI) to the National Healthcare Safety Network (NHSN) as it meets the criteria for a nonculture based test (NCT). Hence, a T2Candida positive result even in the absence of a positive blood culture could be considered for determination of a CLABSI. Responding to concerns from reporting healthcare institutions, NHSN revised reporting criteria starting January 1, 2020: If a NCT is positive and the blood culture is negative 2 days before or 1 day after, the NCT result is not reported [48].

## 10. Conclusions

Invasive candidiasis remains a serious health issue with high mortality especially in patients who do not receive timely and appropriate antifungal therapy. Blood cultures remain the standard and are needed for species identification and susceptibility testing. However, blood cultures are limited by low sensitivity and long turnaround time. Tissue and fluid cultures have similar limitations and often require invasive procedures to obtain samples. T2Candida is a rapid and accurate non-culture-based assay using whole blood directly to diagnose candidemia. In settings where candidemia is highly prevalent, T2Candida could be incorporated into best practice treatment guidelines in conjunction with blood cultures to guide management of patients with suspected invasive candidiasis [49]. Its high negative predictive value makes it a valuable antifungal stewardship asset for clinicians to confidently stop or de-escalate antifungal therapy. Limitations include the inability to detect *Candida* outside of five major species and low sensitivity among populations with a low the prevalence of invasive candidiasis. Further studies in patients with invasive candidiasis are needed to validate and quantify the effect of T2Candida-based management strategies on outcomes, including mortality, length of stay, and duration of antifungal therapy.

## Figures and Tables

**Figure 1 jof-07-00178-f001:**
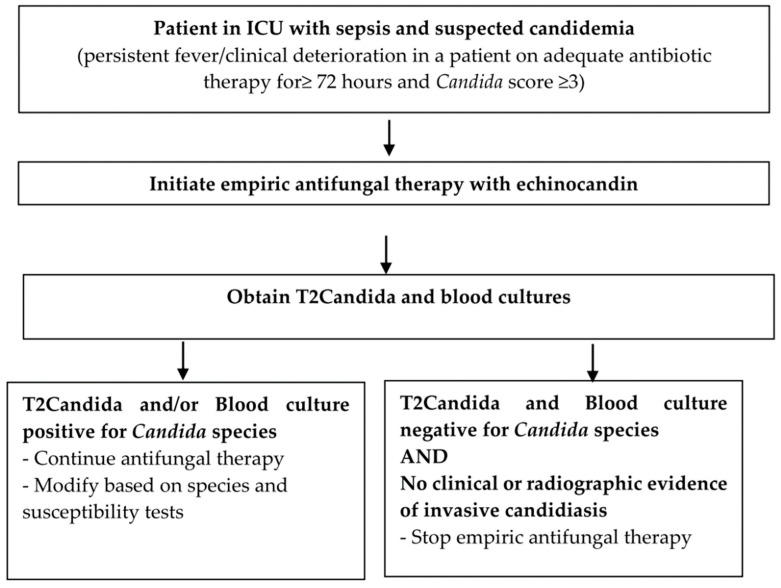
Flowchart for management of suspected candidemia. Note: Candida score defining presumptive candidiasis as Candida score ≥ 3. Candida score component and points: Severe sepsis–2, Multifocal Candida colonization–1, Total parenteral nutrition–1, Surgery on ICU admission–1.

**Table 1 jof-07-00178-t001:** Sensitivity and specificity of T2Candida for detection of *Candida* species in seeded whole blood [9].

*Candida* Species	No. Samples Tested	Sensitivity %	Specificity %
*C. albicans/C. tropicalis*	39	100	98.2
*C. parapsilosis*	18	100	95.8
*C. glabrata/C. krusei*	33	100	100
Combined	90	100	97.8

**Table 2 jof-07-00178-t002:** Time to detection of *Candida* species by T2MR and blood cultures in seeded whole blood [9].

*Candida* Species	Test Platform *	No. Samples Tested (% Positive)	Median Time to Detection in Hours (± SD)	*p*-Value
*C. albicans*	Blood culture	20 (100)	106 ± 5.26	<0.001
	T2MR	20 (100)	3.85 ± 0.29	
*C. tropicalis*	Blood culture	20 (100)	30.58 ± 2.13	<0.001
	T2MR	13 (100)	3.57 ± 0.32	
*C. parapsilosis*	Blood culture	20 (100)	78.25 ± 4.46	<0.001
	T2MR	18 (100)	3.6 ± 0.3	
*C. glabrata*	Blood culture	20 (0)	NA (no growth by day7)	NA
	T2MR	20 (100)	3.6 ± 0.27	
*C. krusei*	Blood culture	20 (100)	40.5 ± 2.23	<0.001
	T2MR	19 (100)	3.83 ± 0.27	

* N = 16 per concentration.

**Table 3 jof-07-00178-t003:** Agreement of T2MR and blood cultures for detection of *Candida* species in spiked human whole blood [10].

		Blood Culture		Total	T2MR Agreement with Blood Culture
		Positive (*N* = 90)	Negative (*N* = 43)	*N* = 133	
**T2MR Candida**	Positive	88	0		Positive 97.8%
	Negative	2	43		Negative 100%

**Table 4 jof-07-00178-t004:** Limit of detection of *Candida* species in spiked human whole blood by T2MR [10].

CFU/mL *	Detection of Specific *Candida* Species				
	*C. albicans*	*C. tropicalis*	*C. parapsilosis*	*C. glabrata*	*C. krusei*
1 CFU/ml	93.8%	75%	100%	93.8%	81.3
2 CFU/ml	93.8%	87.5%	100%	100%	100%
3 CFU/ml	100%	100%	100%	100%	100%
Limit of detection	3 CFU/ml	3 CFU/ml	1 CFU/ml	2 CFU/ml	2 CFU/ml

* *N* = 16 per concentration.

**Table 5 jof-07-00178-t005:** Summary of clinical trials of T2Candida for the diagnosis of candidemia.

Author Year	Type of Study	No. of Subjects and Study Population	Sensitivity %	Specificity %	Limit of Detection CFU/mL	Time to Result (hours)	Comments and Limitations
Mylonakis 2015 USA [14]	Multicenter, Prospective. Compared T2 and BC for detection of candidemia (DIRECT)	(a) 1801 hospitalized adult patients that had BC done as standard of care (b) 250 patient whole blood samples were spiked with <1 to 100 CFU of 5 Candida species detected by T2MR	Overall: 91.1% CA/CT: 92.3% CP: 94.2% CG/CK: 88.1%	Overall: 99.4% CA/CT: 98.9% CP: 99.9% CG/CK: 99.9%	CT&CK: 1 CA&CG: 2 CP: 3	4.4 ± 1 (positive result) 4.2 ± 0.9 (negative result)	-Indeterminate T2 result in 245 cases -Only 6 patients had candidemia therefore sensitivity analysis is primarily based on spiked blood samples -Estimated PPV 32.6% to 98% for disease prevalence of 1% to 50% -Estimated NPV 99.9% to 91.6% for disease prevalence of 1–50%
Clancy 2018 USA [15]	Multicenter Prospective Comparison of T2 and BC in patients with confirmed candidemia (DIRECT2)	152 hospitalized adult patients that had BC confirmed candidemia with *Candida* species: CA, CT, CP, CG, and CK. Follow up BC and concurrent T2 were collected	Overall: 89% -T2 positive in 45% (69/152) -BC positive in 24% (36/152)	NA	NA	NA	-Median time from initial candidemia to collection of follow up BC and concurrent T2 was 55.5 h -At enrollment 74% were on antifungal therapy -T2 positive/BC negative in 24% (37/152) patients and was significantly associated with neutropenia, prior antifungals, and *C. albicans* candidemia

T2 = T2Candida; BC= blood culture; CFU=colony forming units; CA = *C. albicans*; CT = *C. tropicalis*; CP = *C. parapsilosis*; CG = *C. glabrata*; CK = *C. krusei*; PPV = positive predictive value; NPV = negative predictive value.

## Data Availability

Not applicable.

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
