# Peer review of "T2Candida for the Diagnosis and Management of Invasive Candida Infections"

_jof, 2021, doi:10.3390/jof7030178_

Round 1

Reviewer 1 Report

The current review manuscript, entitled"  T2Candida for the diagnosis and management of invasive Candida infections” discuss the preclinical and clinical studies of T2Candida for diagnosis of candidemia and review the current literature on its use in deep-seated candidiasis, role in patient management and prognosis, clinical utility in unique populations and nonblood specimens, and as an antifungal stewardship tool. In addition, they summarized the strengths and limitations of this promising non-culture-based diagnostic test.

The data is interesting and the configuration and presentation of the data are attractive. The manuscript is well written, the tables and graph are perfect, the methodology is excellent, and the corresponding researcher and coauthors have experienced. The manuscript requires minor revisions with confirmation or proven for my own questions.

It is true that the incidence of infections due to Candida has increased and is certainly often more highly associated with certain types of infections. However, what is not discussed in the manuscript is how treatment/ management outcomes are improved by identifying the correct species. By comparing the different methods, the data is convincing that there is good evidence for T2Candida.

How about in mixed candidemia? Do you have any concepts?

  • Medina N, et al. MixInYeast: A Multicenter Study on Mixed Yeast Infections. J Fungi (Basel). 2020 Dec 29;7(1):13. doi: 10.3390/jof7010013. PMID: 33383783; PMCID: PMC7823447.

Conclusions are very weak and repetitive; it is not clear what the base for these conclusions is?

In the end, it is unclear how this information is to be used. What do the authors suggest?

Therefore newly published references have been missed, update references

Author Response

We thank the reviewer for their time and consideration to help improve the quality of our work.  All suggestion and comments are addressed below.

Reviewer Comment 1: The manuscript requires minor revisions with confirmation or proven for my own questions. It is true that the incidence of infections due to Candida has increased and is certainly often more highly associated with certain types of infections. However, what is not discussed in the manuscript is how treatment/ management outcomes are improved by identifying the correct species. By comparing the different methods, the data is convincing that there is good evidence for T2Candida.

Response: This is an important issue that the reviewer makes. Rapid identification of the species and sooner initiation of appropriate antifungal therapy would be expected to improve outcomes, however current studies have not demonstrated a significant mortality benefit. This has been addressed in the limitations section (lines 385-388) of the manuscript.

Reviewer Comment 2: How about in mixed candidemia? Do you have any concepts?

Response: At this time, the grouped reporting of the T2candida panel as C. albicans/ tropicalis and C. Glabrata/ Krusei makes it difficult to differentiate mixed infections. This has been noted and the suggested reference has been included around lines 389-391 in the limitations section.

Reviewer Comment 3: Therefore newly published references have been missed, update references

Response: This newly published study by Medina et al on mixed candidemia has been added as reference 47 and the list updated as appropriate.

Reviewer Comment 4: Conclusions are very weak and repetitive; it is not clear what the base for these conclusions is? In the end, it is unclear how this information is to be used. What do the authors suggest?

Response: The conclusion has been edited to better reflect the potential utility and clinical application of theT2Candida panel as to how this information can be used, it limitations, and future areas of research. (highlighted sections).

Reviewer 2 Report

This is an excellent review that summarizes the role of T2Candida in the diagnosis and management of invasive candida infections. However, a few points may need to be reconsidered/reviewed:

Line 28: The crude mortality rates are much wider than cited

Lines 71-72: For automated blood culture bottles, the actual sensitivity needs to be provided and not '100% except C glabrata'

Lines 74-75: The emphasis on the rapidity of results is missing. 

Line 183: This is the issue that needs a bit more discussion. C. kefyr was missed because it is not included in the panel. In this context it should be emphasized in the review that blood cultures are still crucial although once the etiology is ascertained (one of the 5 common species), then follow up T2 Candida may have some role 

Line 213: The cost saving element has some limitations as the equation is ever changing. The cost of antifungals is much more affordable now so what used to be cost effective is not necessarily cost effective. 

Line 414: Figure 1 has a problem. The suggestion is that T2Candida and/or Blood cultures are negative leads to cessation of therapy. Surely, this should be "and" and not "and/or" because if one of them is not negative, then it is positive.

General coment: Authors should consider adding a short paragraph in relation to the potential inclusion of T2Candida in the good practice reommendations (in relation to candidaemia) e.g. ECMM score 

Author Response

We thank the reviewer for their time and consideration to help improve the quality of our work.  All points have addressed below.

Reviewer Comment 1: Line 28: The crude mortality rates are much wider than cited

Response: Due to the wide variation in mortality rates depending on the source and publication, this line has been edited to state that invasive candidiasis has significant attributable mortality without giving a specific percentage (see lines 28-29).

Reviewer Comment 2: Lines 71-72: For automated blood culture bottles, the actual sensitivity needs to be provided and not '100% except C glabrata.

Response: The reference was revisited and study information about the growth detection at 5 days is included to make the reporting more specific as requested. (See line 71).

Reviewer Comment 3: Lines 74-75: The emphasis on the rapidity of results is missing.

Response: Emphasis was added to the rapidity of results (see line 75).

Reviewer Comment 4: Line 183: This is the issue that needs a bit more discussion. C. kefyr was missed because it is not included in the panel. In this context it should be emphasized in the review that blood cultures are still crucial although once the etiology is ascertained (one of the 5 common species), then follow up T2 Candida may have some role.

Response: The lack of certain species identification is an important point on the limitations of T2Candida. Based on this feedback emphasis was placed in the conclusion section that T2Candida does not replace the need to perform blood cultures in patients with suspected invasive candidiasis (See line 432-433).

Reviewer Comment 5: Line 213: The cost saving element has some limitations as the equation is ever changing. The cost of antifungals is much more affordable now so what used to be cost effective is not necessarily cost effective.

Response: The flux in costs of antifungals over time is a limitation to all cost savings studies and an important point we agree needs to be noted when evaluating such studies. The text has been modified to reflect this (line 232).

Reviewer Comment 6: Line 414: Figure 1 has a problem. The suggestion is that T2Candida and/or Blood cultures are negative leads to cessation of therapy. Surely, this should be "and" and not "and/or" because if one of them is not negative, then it is positive.

Response: Thank you for catching this error. The text has been corrected to state “and”.

Reviewer Comment 7: General comment: Authors should consider adding a short paragraph in relation to the potential inclusion of T2Candida in the good practice recommendations (in relation to candidaemia) e.g. ECMM score

Response: In reworking the conclusion section per the alternative reviewer’s recommendations, we have added short point about the utility of this test in clinical practice. Specifically, the ability incorporate T2Candida into institution treatment guidelines (See lines 437-439) and reference has been added to the ECMM score as a best clinical practice.